# Antibiotic consumption and utilization at a large tertiary care level hospital in Uganda: A point prevalence survey

**Mark Kizito** [1,2☯] *, **Darius Owachi** [2☯], **Falisy Lule** [2☯], **Laura Jung** [3☯], **Vivian Bazanye** [4‡], **Ibrahim Mugerwa** [5‡], **Susan Nabadda** [5‡], **Charles Kabugo** [2‡]

**1** Department of Internal Medicine, School of Health Sciences, Soroti University, Soroti, Uganda, **2** Kiruddu National Referral Hospital, Kampala, Uganda, **3** Leipzig University Medical Center, Division of Infectious Diseases and Tropical Medicine, Leipzig, Germany, **4** Infectious Diseases Institute, Makerere University, Kampala, Uganda, **5** Department of National Health Laboratories and Diagnostic Services, Ministry of Health, Kampala, Uganda

☯ These authors contributed equally to this work.
‡ These authors also contributed equally to this work
* mkizito2@gmail.com

**Data Availability Statement:** All relevant data are within the paper and its Supporting Information files.

## Abstract

### Introduction

Effective antimicrobial stewardship programs require data on antimicrobial consumption (AMC) and utilization (AMU) to guide interventions. However, such data is often scarce in low-resource settings. We describe the consumption and utilization of antibiotics at a large tertiary-level hospital in Uganda.

### Methods

In this cross-sectional study at Kiruddu National Referral Hospital, we analyzed medicine delivery records for the period July 2021 to June 2022, accessed on 12/08/2022, to extract AMC data expressed as defined daily doses (DDDs) per 1000 inhabitants per day (DID). We used the WHO point prevalence survey (PPS) to analyze AMU data with a systematic sampling of outpatient department (OPD) for a period between June and August 2022 and selected all inpatient department (IPD) patients admitted before 8:00 AM on 27/11/2022. AMU data was analyzed as the proportion of individual antibiotic prescriptions, indications for prescriptions, and compliance with the national treatment guidelines. Both AMC and AMU data were categorized by the WHO AWaRe (access, watch, and reserve) criteria.

### Results

In the year 2021–2022, a total of 6.05 DID of antibiotics were consumed comprising 3.61 DID (59.6%) access, 2.44 DID (40.3%) watch, and 0.003 (0.1%) reserve antibiotics. The most consumed antibiotics comprised penicillin (1.61 DID, 26.7%), cephalosporins (1.51, 25%), and imidazole (1.10 DID, 18.1%).

A total of 119/211 (56%) patients in the OPD and 99/172 (57.5%) patients in the IPD were prescribed antibiotics. Of the 158 OPD antibiotic prescriptions, 73 (46.2%) were

**Funding:** This study was funded through collaborations with the Surveillance Partnership to Improve Data for Action on Antimicrobial Resistance (SPIDAAR) program [funded by the Pfizer/Wellcome Trust foundation and the Education and Stewardship to Improve Knowledge, Attitude, and Practices towards Antimicrobial Resistance (ESKAPE) project [funded through the GIZ GmbH]. The funder had no role in the design, investigation, analysis, and reporting of this study findings.

**Competing interests:** No authors have competing interests

**Abbreviations:** AMR, Antimicrobial Resistance; AMS, Antimicrobial Stewardship; AMU, Antimicrobial Utilization; ASP, Antibiotic Stewardship Program; AwaRe, Access, Watch, Reserve; CDC, Centers for Disease Control and Prevention; CKD, Chronic Kidney Disease; CVD, Cardiovascular Disease; DDD, Defined Daily Dose; DID, Defined Daily Doses per 1000 inhabitants per day; HIV, Human Immunodeficiency Virus; IV, Intravenous; GIT, Gastrointestinal; IPD, In-patient Department; KNRH, Kiruddu National Referral Hospital; ODK, Open Data Kit; OPD, Outpatient Department; PPS, Point Prevalence Survey; RTI, Respiratory tract infection; USA, United States of America; UCG, Uganda Clinical Guidelines; UTI, Urinary tract infection; WHO, World Health Organization.

access, 72 (45.6%) were watch, 0 (0%) were reserve, and 13 (8.2%) were unclassified antibiotics. Of the 162 IPD antibiotic prescriptions, 62 (38.3%) were access, 88 (54.3%) were watch, 01 (0.6%) was reserve, and 11 (6.8%) were unclassified antibiotics. Indications for antibiotic prescriptions in the OPD comprised respiratory tract infections (53, 38.1%), urinary tract infections (34, 26.6%), gastrointestinal infections (20, 14.4%), sepsis (17, 12.2%), and medical prophylaxis (12, 8.7%). The indications for antibiotic prescriptions in the IPD comprised sepsis (28.2%), respiratory tract infections (18.3%), burn wounds (14.1%), and gastrointestinal infections (14.1%).

## Conclusion

Prescription of watch antibiotics in both OPD and IPD hospital settings was high. Establishment of robust antimicrobial stewardship measures could help improve the rational prescription of antibiotics.

## Introduction

Antimicrobial Resistance (AMR) is a growing silent pandemic that threatens all spheres of human health, including biomedicine, behavioral science, and environmental protection [1]. The global burden of bacterial antibiotic resistance in 2019 was estimated at 1.27 million attributable deaths and 4.95 million associated deaths, with the greatest proportion affecting the Sub-Saharan African region [2]. The negative effects of AMR such as increased healthcare costs, increased length of hospital stays, and increased mortality rates [2–5] are estimated to cause a loss of over $100 trillion in the global economy between 2000 and 2050, disproportionately affecting low- and middle-income countries [4,6,7]. Throughout East Africa, there is a high burden of community acquired infectious disease and increasing rates of AMR [8–10].

In Uganda, bacterial infections, such as pneumonia, TB, and sepsis, accounted for 18.4% of hospital-based mortality in the fiscal year 2014/2015 [11], and high rates of antibiotic resistance have been reported [12–14]. Recent studies showed high rates of antibiotic resistance among patients hospitalized with post-traumatic or burn wound infections in two large tertiary hospitals in Uganda [15,16].

Numerous factors contribute to the development and escalation of AMR including antibiotic use and misuse such as self-medication practices and antibiotic over-prescription, weak AMR diagnostic stewardship systems, non-availability of the right antibiotic formularies for optimal access, and incomplete antibiotic treatment by patients [17–19]. Addressing these factors has been shown to reduce the spread of AMR [20]. The June 2021 G7 meeting's health agenda focused heavily on addressing the then-ongoing COVID-19 pandemic through vaccine equity, preparing for future pandemics, reforming the World Health Organization (WHO), and combating global health threats like antimicrobial resistance where the importance of optimizing the use of antibiotics as a tool to combat AMR and improve infection management was emphasized [1].

Antibiotic consumption and utilization remain critical challenges in low- and middle-income countries (LMICs), where healthcare systems often grapple with limited resources, inadequate regulations, and varying levels of healthcare access. The overuse of antibiotics is a concerning issue, with high rates of antibiotic prescription reported across several countries yet sparsity of data on consumption and utilization persists [21–24]. Few studies have been

conducted to assess antibiotic consumption in Uganda, and a significant gap persists in understanding antibiotic stewardship within tertiary care settings [25]. Tertiary hospitals, due to their role as referral centers for more complex cases, are likely to exhibit a higher use of reserve antibiotics, which are critical in managing multidrug-resistant infections.

The majority of consumption analyses rely on aggregate data from antimicrobial warehouse supplies as proxies for consumption [26,27]. These studies provide valuable insight but fall short of reflecting actual prescribing practices at the point of care in health facilities. However, detailed, facility-level data on antibiotic prescribing behavior, builds the foundation for targeted interventions aimed at optimizing antibiotic use and curbing AMR.

It is therefore imperative to develop effective strategies to curb this growing burden of AMR that poses significant public health and socio-economic challenges. Antimicrobial stewardship measures such as the use of antibiograms have been shown to effectively improve antibiotic prescribing in acute-care settings thus reducing the spread of AMR [28]. However, to inform effective policies and interventions that optimize use and promote equitable access to medicines, it is essential to document trends on antibiotic consumption and utilization in all settings [29]. This study aims to examine antibiotic prescribing practices by determining the prevalence of antimicrobial consumption and utilization specifically within a tertiary care setting in Uganda. By doing so, it seeks to provide more nuanced data on the patterns of antibiotic utilization, particularly the use of reserve antibiotics, and underscore the importance of antibiotic stewardship in these critical healthcare environments.

## Materials and methods

### Study setting and design

This cross-sectional study was conducted at Kiruddu National Referral Hospital (KNRH), a 200-bed capacity tertiary-level public healthcare facility located in the southern suburb region of the capital city of Kampala, Uganda (coordinates 0˚14'54.8"N 32˚36'45.3" E). Established in 2016, the hospital provides both medical and surgical in-patient and outpatient department services, diagnostic services, and the largest renal replacement therapy unit in the country. Being a tertiary-level health facility, the hospital serves the catchment population within the capital city of Kampala, as well as receiving referrals from within the country. Anecdotal hospital data indicates that approximately 100,000 patient visits are made to KNRH each year [30]. The hospital laboratory offers a range of diagnostic services including accredited microbiology and antimicrobial susceptibility testing [31]. The Hospital Pharmacy services are coordinated through the hospital pharmacist with nine pharmacy technicians (Pharmacy Diploma holders) who quantify and forecast for medicines that are primarily obtained from the National warehouse; National Medical Stores. These are delivered on a monthly basis and are provided in 6 pharmacies within the hospital two of which operate 24 hours. Drugs provided include a wide range of Non-Communicable Diseases commodities; Diabetes, Hypertension medicines, and antimicrobial agents. Antimicrobial agents supplied to the hospital are largely guided by annual quantification based on output consumptions of previous years and needs by specialists. All these are provided at no cost to the patient. Rational medicines use is coordinated through the Medicines Therapeutics Committee with sub committees on Infection Prevention and Control, Supply Chain, Antimicrobial Stewardship and Pharmacovigilance [30].

In 2021, KNRH established an Antimicrobial stewardship committee to develop and monitor antimicrobial stewardship activities at the hospital. Among the implemented activities included strengthening the antimicrobial resistance surveillance capacity of the institution as well as assessing antimicrobial consumption and utilization. In this study, we determined the

consumption and utilization of antimicrobials for the fiscal year 2021–2022 which serves as a baseline to monitor the impact of antimicrobial stewardship programs at KNRH.

## Data collection

**Antimicrobial consumption.** All data was accessed on the 12[th] of August 2022 and was fully anonymized before patient records were accessed.

KNRH receives a cyclic supply of medicines through the National Medical Stores (NMS), the official government entity that procures and supplies medicines to all public health facilities in Uganda [32]. We reviewed invoices and delivery logs for medicines supplied to KNRH by NMS for the fiscal year running from 1st July 2021 through 30th June 2022. Data on quantities and types of antibiotics delivered and their respective prices were extracted and analyzed (S1 File).

Antibiotic consumption data was presented as the defined daily dose (DDD) which was calculated by dividing the total grams of the antibiotic consumed by the standard [18]. The DDDs per 1000 inhabitants per day (DID) were calculated by multiplying the DDDs for the antibiotic by 1000 and dividing the product by the Ugandan population for the year 2022 estimated at 44,200,000 [33]. The antibiotics were subcategorized by individual antibiotics, by the WHO AWaRe system, and by the total expenditure of individual and all antibiotics purchased in the fiscal year 2021–2022 [34].

**Antimicrobial utilization.** We reviewed patient records and registers from the outpatient (OPD) and inpatient (IPD) departments of the hospital to extract data on antibiotic utilization. Antibiotic utilization was analyzed as the proportion of patients in whom an antibiotic was prescribed, the type of antibiotics prescribed, the indication (diagnosis) for the antibiotic prescriptions, the proportion of missed doses, and the compliance with the national treatment guidelines [35].

To assess antimicrobial utilization in the outpatient setting, we considered a sample size of 200 patient encounters. The WHO PPS methodology guides that for small-size hospitals with a bed capacity of 200 or under, the sample size can be approximated to the bed capacity of the facility thus the sample size of 200 [29]. A total of 8,923 patients visited the general OPD between 1st June to 31st August 2022. We employed systematic random sampling by selecting every 44th prescription from a total of 8923 prescriptions recorded between 1st June through 31st August 2022, until reaching a desired sample size of 200 prescriptions. For inpatient data, given the small number of hospital beds at 200, we reviewed all records of patients admitted at the hospital before 8:00 AM of the day of the survey, as guided by the WHO point prevalence survey guidelines, to obtain the recommended sample size [29]. The IPD survey was conducted on the 27[th] of November 2022. For both OPD and IPD data, there were no specific exclusion criteria. All patient records that were sampled were analyzed.

Data on antibiotic prescriptions, indications, dosages, routes of administration, individual wards, missed doses, and medical history were extracted using an open data kit (ODK Collect v2022.4.3) tool designed based on the WHO point prevalence survey methodology. No variables were removed or changed to suit our context.

Infectious diagnoses were classified as a) respiratory tract infections (to include sinusitis, bronchitis, pharyngitis, and/or pneumonia); b) gastrointestinal tract infections (to include gastritis, enterocolitis, appendicitis); c) urinary tract infections (to include uncomplicated or complicated urinary tract infections, pyelonephritis, cystitis, and urethritis), d) central nervous system infections (to include meningitis, encephalitis); skin and soft tissue infections (to include cellulitis, skin abscesses, burns wounds etc.); malaria (where microbiologic confirmation of malaria was made); tuberculosis (where microbiologic confirmation of TB bacilli was

made or patient was taking anti-TB medication); and sepsis (bloodstream infection with evidence of organ dysfunction).

### Data analysis and management

We coded and double-entered data into Microsoft Excel to remove any inconsistencies and analyzed using descriptive statistics such as frequencies.

### Ethical consideration

Ethical approval for the study was obtained from the Uganda National Health Laboratory Services Research and Ethics Committee under reference SN_UNHL-2020-9 and registered with the Uganda National Council for Science and Technology, under reference **HS1268ES.** Permission to review patient records was sought from the hospital administration of KNRH. The requirement for Informed consent was waived by the IRB and all data was fully anonymized before patient records were accessed. All data were stored at a secure server located at the Infectious Diseases Institute, Makerere University.

## Results

### Characteristics of study patients

A total of 211 patients in the OPD and 172 patients in the IPD were included in the study. Females comprised 130 (61.6%) in the OPD and 100 (58.1%) in the IPD. The median (IQR) age was 32 (15–48) years in the OPD and 40 (26–60.5) years in the IPD.

Infectious diseases were the most common diagnoses encountered in both outpatient (150, 82%) and in-patient departments (67, 97%). Among the comorbidities, hypertension (23, 48.9%), diabetes mellitus (7, 14.9%), and liver disease (5, 10.6%) were the most common among outpatients while HIV (21, 44.7%), liver disease (7, 14.7%), and chronic kidney disease (5, 10.6%) were prevalent among admitted patients (**Table 1**).

### Antimicrobial consumption and expenditure

A total of 267,595 DDDs of antimicrobials were consumed in the year 2021/2022 at KNRH with a total DID of 6.05 DDDs per 1,000 inhabitants per day. By the AWaRe classification, 3.61 (59.6%) DID were access, 2.44 (40.3%) DID were watch and 0.003 (0.1%) DID were reserve antibiotics. By antibiotic classes, the most consumed antibiotics comprised penicillin 1.61 DID (26.7%), cephalosporins 1.51 DID (25%), and imidazole 1.10 (18.1%); all collectively comprising more than two-thirds of the total antibiotics consumed in the year 2021–2022 (**S1 File**). Of the individual antibiotics, metronidazole (1.19 DID, 19.6%), ceftriaxone (0.83 DID, 13.6%), amoxicillin-clavulanate (0.70 DID, 11.5%), cefixime (0.68 DID, 11.2%), and amoxicillin (0.68 DID, 11.2%), collectively contributed to 4.08 DID (67.2%) of total antibiotics consumed (**Table 2**).

In the fiscal year 2021–2022, total expenditure on antibiotics was $110,785, which comprised 16% of the total hospital budget expenditure of $682,985. The bulk of antibiotic expenditure was on watch antibiotics costing $66,527 (60%) and access antibiotics costing $35,029 (31.6%) while the least expenditure was on reserve antibiotics at $3,939 (3.6%) and unclassified antibiotics at $5,290 (4.8%). By individual drugs, the most expenditure was on ceftriaxone $31,094 (28.1%), metronidazole $11,483 (10.4%), levofloxacin $10,880 (9.8%), and amoxicillin-clavulanic acid $9,722 (8.8%) (**Table 2**).

**Table 1. Characteristics of study patients.**

| Variable | OPD n (%) | IPD n (%) |
|---|---|---|
| Total Patients | 211 | 172 |
| Sex, Female | 130 (61.6) | 100 (58.1) |
| Age, Median (Interquartile range) | 32 (15–48) | 40 (26–60.5) |
| **Antibiotic Therapy** | | |
| Patients on Antibiotics | 119 (56.4) | 99 (57.6) |
| Total number of Antibiotics Prescribed | 158 | 162 |
| Antibiotics per encounter | 1.33 | 1.64 |
| **Diagnoses** | | |
| Burn Wounds | 0 (0) | 10 (14.5) |
| Central nervous system/Meningitis infections | 0 (0) | 3 (4.3) |
| Gastrointestinal Infection | 20 (10.9) | 7 (10.1) |
| Malaria | 13 (7.1) | 2 (2.9) |
| Peptic ulcer disease | 17 (9.3) | 2 (2.9) |
| Peripheral neuropathy | 16 (8.7) | 0 (0) |
| Pulmonary tuberculosis | 2 (1.1) | 3 (4.3) |
| Respiratory tract infection | 53 (29.0) | 13 (18.8) |
| Sepsis | 17 (9.3) | 14 (20.3) |
| Skin infections | 11 (6.0) | 9 (13.0) |
| Urinary Tract Infection | 34 (18.6) | 6 (8.7) |
| Total | 183 (100) | 69 (100) |
| **Comorbidities** | | |
| Asthma | 4 (8.5) | 2 (4.3) |
| Chronic kidney disease | 1 (2.1) | 5 (10.6) |
| Chronic obstructive pulmonary disease | 3 (6.4) | 3 (6.4) |
| Cardiovascular disease (Stroke, heart disease) | 3 (6.4) | 3 (6.4) |
| Diabetes Mellitus | 7 (14.9) | 3 (6.4) |
| HIV | 1 (2.1) | 21 (44.7) |
| Hypertension | 23 (48.9) | 3 (6.4) |
| Liver disease | 5 (10.6) | 7 (14.9) |
| Total | 47 (100) | 47 (100) |

OPD–outpatient department, IPD–inpatient department.

## Antibiotic utilization in the outpatient department

Of the 211 patients records reviewed from the OPD, 119 (56%) were prescribed antibiotics. A total of 158 antibiotics were prescribed, with an average of 1.33 antibiotic prescriptions per encounter (158/119). However, only 35/119 (29%) of the patients received appropriate antibiotic prescriptions in accordance with the national treatment guidelines (**S2 File**).

Of the 158 antibiotic prescriptions, 73 (46.2%) comprised access antibiotics while 72 (45.6%) comprised watch antibiotics. No reserve antibiotics were prescribed in the OPD. By antibiotic classes, the most prescribed antibiotics included penicillin (44, 27.8%), cephalosporins (40, 25.3%), and imidazole (29, 18.4%). By individual antibiotics, cefixime (33, 20.9%), metronidazole (28, 17.7%), amoxicillin-clavulanate (14, 8.9%), and azithromycin (13, 8.2%) were the most prescribed antibiotics (**S3 File**).

The common indications for antibiotic prescriptions in the OPD included respiratory tract infections (53, 38.1%), urinary tract infections (34, 26.6%), gastrointestinal infections (20, 14.4%), and sepsis (17, 12.2%) (**Table 3**).

**Table 2. Antimicrobial consumption and expenditure during the fiscal year 2021/2022.**

| Antimicrobial consumption | DDD (%) | DID | Expenditure* (%) |
|---|---|---|---|
| **AWaRe classification** | | | |
| Access | 159,599 (59.6) | 3.61 | 35,029 (31.6) |
| Watch | 107,846 (40.3) | 2.44 | 66,527 (60.1) |
| Reserve | 150 (0.1) | 0.00 | 3,939 (3.6) |
| Unclassified | 0 (0) | 0 | 5,290 (4.8) |
| Total | 267,595 (100) | 6.05 | 110,785 (100) |
| **Antibiotic class** | | 0 | |
| Penicillin | 71,308 (26.6) | 1.61 | 25,797 (23.3) |
| Cephalosporins | 66,800 (25.0) | 1.51 | 38,263 (34.5) |
| Imidazole | 48,501 (18.1) | 1.10 | 11,483 (10.4) |
| Tetracyclines | 25,000 (9.3) | 0.57 | 473 (0.4) |
| Fluoroquinolones | 23,885 (8.9) | 0.54 | 12,933 (11.7) |
| Macrolides | 19,840 (7.4) | 0.45 | 5,800 (5.2) |
| Sulfonamides | 10,890 (4.1) | 0.25 | 0 (0) |
| Carbapenems | 625 (0.2) | 0.01 | 11,806 (10.7) |
| Aminoglycosides | 600 (0.2) | 0.01 | 2,565 (2.3) |
| Glycopeptides | 146 (0.1) | 0.00 | 1,664 (1.5) |
| Total | 267,595 (100) | 6.05 | 110,785 (100) |
| **Individual Antibiotics** | | 0 | |
| Amikacin | 600 (0.2) | 0.01 | 2,565 (2.3) |
| Amoxicillin | 30,000 (11.2) | 0.68 | 1,984 (1.8) |
| Amoxicillin-Clavulanate | 30,858 (11.5) | 0.70 | 9,722 (8.8) |
| Ampicillin-Cloxacillin | 8,062 (3.0) | 0.18 | 1,591 (1.4) |
| Azithromycin | 12,500 (4.7) | 0.28 | 1,040 (0.9) |
| Cefixime | 30,000 (11.2) | 0.68 | 6,219 (5.6) |
| Cefotaxime | 301 (0.1) | 0.01 | 950 (0.9) |
| Ceftriaxone | 36,500 (13.6) | 0.83 | 31,094 (28.1) |
| Ciprofloxacin | 13,020 (4.9) | 0.29 | 2,053 (1.9) |
| Clarithromycin | 5,040 (1.9) | 0.11 | 4,412 (4.0) |
| Cotrimoxazole | 10,890 (4.1) | 0.25 | 0 (0) |
| Doxycycline | 25,000 (9.3) | 0.57 | 473 (0.4) |
| Erythromycin | 3,000 (1.1) | 0.07 | 348 (0.3) |
| Flucloxacillin-Amoxicillin | 0 (0) | 0 | 5,290 (4.8) |
| Imipenem | 150 (0.1) | 0.00 | 3,939 (3.6) |
| Levofloxacin | 6,865 (2.6) | 0.16 | 10,880 (9.8) |
| Meropenem | 475 (0.2) | 0.01 | 7,867 (7.1) |
| Metronidazole | 52,501 (19.6) | 1.19 | 11,483 (10.4) |
| Penicillin-Benzathine | 980 (0.4) | 0.02 | 724 (0.7) |
| Piperacillin-Tazobactam | 707 (0.3) | 0.02 | 6,487 (5.9) |
| Vancomycin | 146 (0.1) | 0.00 | 1,664 (1.5) |
| Total | 267,595 (100) | 6.05 | 110,785 (100) |

DDD: Defined daily dose, DID: DDD per 1000 inhabitants per day, *Expenditure in US Dollars.

## Antibiotic utilization in the In-patient Department (IPD)

Of the 172 patients who were reviewed in the IPD, 99 (57.5%) were prescribed antibiotics. A total of 162 antibiotics were prescribed with an average of 1.63 antibiotic prescriptions per

**Table 3. Antibiotic utilization by AWARE classification, antibiotic class, prescribed antibiotics, and indications for antibiotic use.**

| Antibiotic Utilization | OPD n (%) | IPD n (%) |
|---|---|---|
| **Utilization by AWaRe Classification** | | |
| Access | 73 (46.2) | 62 (38.3) |
| Watch | 72 (45.6) | 88 (54.3) |
| Reserve | 0 (0) | 1 (0.6) |
| Unclassified | 13 (8.2) | 11 (6.8) |
| Total | 158 (100) | 162 (100) |
| **Utilization by Antibiotic class** | | |
| Penicillin | 44 (27.8) | 23 (14.2) |
| Cephalosporins | 40 (25.3) | 63 (38.9) |
| Imidazole | 29 (18.4) | 39 (24.1) |
| Macrolides | 18 (11.4) | 3 (1.9) |
| Fluoroquinolones | 15 (9.5) | 19 (11.7) |
| Tetracyclines | 8 (5.1) | 1 (0.6) |
| Sulfonamides | 0 (0) | 10 (6.2) |
| Aminoglycosides | 0 (0) | 2 (1.2) |
| Glycopeptides | 0 (0) | 1 (0.6) |
| Nitrofurantoin | 2 (1.3) | 1 (0.6) |
| Amphenicols | 2 (1.3) | 0 (0) |
| Total | 158 (100) | 162 (100) |
| **Utilization by Individual Antibiotics** | | |
| Amoxicillin | 13 (8.2) | 1 (0.6) |
| Amoxicillin-clavulanate | 14 (8.9) | 2 (1.2) |
| Azithromycin | 13 (8.2) | 3 (1.9) |
| Ampicillin-cloxacillin | 11 (7.0) | 5 (3.1) |
| Benzyl penicillin | 0 (0) | 2 (1.2) |
| Cefixime | 33 (20.9) | 1 (0.6) |
| Cefotaxime | 0 (0) | 2 (1.2) |
| Cefpodoxime | 1 (0.6) | 0 (0) |
| Ceftriaxone | 4 (2.5) | 60 (37.0) |
| Cefuroxime | 2 (1.3) | 0 (0) |
| Chloramphenicol | 2 (1.3) | 0 (0) |
| Ciprofloxacin | 5 (3.2) | 4 (2.5) |
| Clarithromycin | 3 (1.9) | 0 (0) |
| Cotrimoxazole | 0 (0) | 10 (6.2) |
| Doxycycline | 8 (5.1) | 1 (0.6) |
| Erythromycin | 2 (1.3) | 0 (0) |
| Flucloxacillin | 6 (3.8) | 9 (5.6) |
| Gentamycin | 0 (0) | 2 (1.2) |
| Levofloxacin | 10 (6.3) | 15 (9.3) |
| Metronidazole | 28 (17.7) | 39 (24.1) |
| Nifuroxazide | 2 (1.3) | 0 (0) |
| Nitrofurantoin | 0 (0) | 1 (0.6) |
| Piperacillin-tazobactam | 0 (0) | 4 (2.5) |
| Tinidazole | 1 (0.6) | 0 (0) |
| Vancomycin | 0 (0) | 1 (0.6) |
| Total | 158 (100) | 162 (100) |

(*Continued*)

**Table 3.** (Continued)

| Antibiotic Utilization | OPD n (%) | IPD n (%) |
|---|---|---|
| **Indications for antibiotic use** | | |
| Respiratory tract Infection | 53 (38.1) | 13 (18.3) |
| Urinary tract Infection | 37 (26.6) | 6 (8.5) |
| Sepsis | 17 (12.2) | 20 (28.2) |
| Gastrointestinal infection | 20 (14.4) | 10 (14.1) |
| Central nervous system infection | 0 (0) | 3 (4.2) |
| Burn Wound Infection | 0 (0) | 10 (14.1) |
| Medical prophylaxis | 12 (8.6) | 7 (9.9) |
| Surgical prophylaxis | 0 (0) | 2 (2.8) |
| Total | 139 (100) | 71 (100) |

OPD–outpatient department, IPD–inpatient department.

encounter (162/99). However, only 31/99 (31%) of the patients received appropriate antibiotic prescriptions in accordance with the national treatment guidelines (**S4 File**).

By the AWaRe classification, 62 (38.3%) were access, 88 (54.3%) were watch, 01 (0.6%) were reserve while 11 (6.8%) were unclassified antibiotics. By antibiotic classification, the most prescribed antibiotics in the IPD included cephalosporins (63, 38.9%), imidazole (39, 24.1%), and fluoroquinolones (19, 11.7%). By individual antibiotics, ceftriaxone (60, 37%), metronidazole (39, 24.1%), and levofloxacin (15, 9.3%) were the most prescribed.

The common indications for antibiotic prescriptions in the IPD comprised sepsis (20, 28.2%), respiratory tract infections (13, 18.3%), gastrointestinal infections (10, 14.1%), and burn wound infections (10, 14.1%) (**Table 3**).

## Discussion

In this study, we assessed antibiotic consumption and utilization at a large tertiary hospital in Uganda. We found a relatively high rate of antibiotic prescription with more than half of patients receiving antibiotic prescriptions in both OPD and IPD settings. More than one-quarter of a million DDDs of antibiotics was consumed annually, with three-fifths comprising access antibiotics and two-fifths comprising watch antibiotics but negligible reserve antibiotics. The total DDD per 1000 inhabitants per day was 6.05. However, there were more prescriptions of watch antibiotics in the IPD compared to the OPD, and generally low compliance with the national treatment guidelines in both OPD and IPD settings.

### Antimicrobial consumption

The overall consumption of antibiotics at Kiruddu National Referral Hospital in the financial 2021–2022 of 6.05 DID was significantly lower than what was reported as the national average consumption of 29.02 for all antimicrobials for the year 2021 [27]. To the best of our knowledge, this could be one of the first studies to report on antibiotic consumption from a public tertiary care facility in Uganda. However, we believe that this study provides a baseline for measuring trends in antibiotic consumption at KNRH which can be used to evaluate the performance of antimicrobial stewardship programs to reduce antibiotic use.

The most consumed antibiotic classes were penicillin, cephalosporins, and imidazole, which are consistent with other studies done in multiple countries reporting data to WHO GLASS [36]. For example, a South African study reported that penicillin and imidazole were

the most commonly consumed antibiotic classes from 2014 to 2018 [37]. Another study done among 14 Kenyan hospitals also reported similar findings [38]. By contrast, the most consumed antibiotic class in Tanzania was tetracycline [39]. The differences in the Ugandan and Tanzanian national treatment guidelines as well as different procurement patterns in public and private health sectors could explain this observation [27].

In our study, about 60% of the IPD antibiotic consumption were access antibiotics. Other studies conducted in Uganda [33,38], Ethiopia [39], and Tanzania [37] revealed similar findings. The observed antibiotic consumption for access antibiotics at KNRH also compares with the WHO AWaRe recommendation of at least 60% access antibiotics in total antibiotic consumption [40].

The WHO Model List of Essential Medicines classified antibiotics into Access, Watch, and Reserve (AWaRe) categories for the treatment of 31 priority bacterial infections as a tool to facilitate antibiotic stewardship and optimal use. WHO recommended that antibiotics in the Access group be available at all times as treatments for many types of common infections (for example, amoxicillin to treat pneumonia). Watch antibiotics have an increased resistance potential and are recommended as first-choice or second-choice treatments for a small number of infections (for example, ciprofloxacin to treat urinary tract infection). Their use should be monitored for appropriateness as part of routine stewardship activities and reduced to avoid further development of resistance. The third group, Reserve, includes broad-spectrum antibiotics (e.g., colistin) that should be considered as last-resort options and used only in the most severe circumstances when all other alternatives have failed [41]. Therefore, WHO's AWaRe classification promotes narrow-spectrum antibiotics over broad-spectrum ones to curb antibiotic resistance [42,43]. Our study revealed that Watch category antibiotics exceeded recommended levels of 40% [42], comprising 45.6% and 54.3% of overall antibiotic use in OPD and IPD, respectively. This group has a high potential for resistance and its increased use could be a potential driver for AMR in Kiruddu National Referral Hospital. This highlights the urgency for robust antimicrobial stewardship protocols to improve prescription practices. Adherence to AWaRe guidelines can mitigate selection pressure and the spread of resistance, ensuring the effectiveness of antibiotics.

The limited consumption of reserve antibiotics in our study is a predictable outcome due to the public procurement system being centered on primary healthcare [26], as well as the prohibitive cost of reserve antibiotics which makes them unaffordable for a sizable portion of the population. Additionally, Kiruddu National Referral Hospital implemented a policy restricting the dispensing of reserve antibiotics to instances where culture and sensitivity results demonstrate the necessity for such antibiotics. While the overall minimal use of reserve antibiotics is encouraging in terms of antibiotic stewardship, it could also potentially suggest lack of access for patients who may require them in situations where resistance to first- and second-line treatment options emerge [16,44].

## Antimicrobial utilization: High antibiotic prescriptions

We found that one in every two patients visiting the OPD or admitted to the IPD were prescribed antibiotics. The high antibiotic prescription rate observed in this study is higher than the recommended WHO target of 40% [29]. A possible explanation might be due to the high burden of infectious diseases reported in Uganda, resulting in increased risk for hospitalization and mortality [45,46]. However, overuse of antibiotics due to lack of awareness and training in health care staff might also play a role. Our findings also compare with studies done in other public healthcare facilities in Uganda [47,48], as well as other Sub-Saharan African countries where the burden of infectious diseases is well documented [22,23,49]. Unfortunately, the

over-prescription of antibiotics may contribute to the propagation of antibiotic resistance with the Sub-Saharan African region projected to suffer the greatest impact of AMR [2,50].

Several studies reveal increasing antimicrobial resistance to common antibiotics, particularly ceftriaxone, used in the local settings [16,51]. Developing and strengthening of antimicrobial stewardship measures such as the use of antibiograms or adopting evidence-based empirical treatment guidelines could guide clinicians on the rational use of antibiotic prescriptions and help reduce the spread of AMR [28,42,52]. An Antibiogram is a periodic profile of antimicrobial susceptibilities of various organisms isolated from patients within an institution. It is commonly utilized to monitor recent antimicrobial susceptibility patterns in order to guide empirical antimicrobial therapy selection [53].

## Antimicrobial utilization: Indications for antibiotic use

In our study, the commonly prescribed antibiotics in the IPD included ceftriaxone, metronidazole, levofloxacin, and cotrimoxazole while cefixime, metronidazole, amoxicillin-clavulanic acid, and azithromycin were the most prescribed in the OPD. Several studies in Uganda have reported similar patterns of prescription of these antibiotics [15,21,47,48,54], a trend also observed in other countries in the Sub-Saharan region [22,55,56]. One possible explanation for this similarity in prescriptions is the listing of these antimicrobial agents on the WHO list of essential medicines that often guide local treatment guidelines [35,41]. Levofloxacin is often reserved for the treatment of drug-resistant infections such as multi-drug resistant tuberculosis [57], but studies indicate a tendency to inappropriately prescribe it in non-TB infections [58].

In this study, the indications for antibiotic use were respiratory tract infections, urinary tract infections, sepsis, gastrointestinal infections, and burn wound infections. This agrees with what has been reported by other studies done in Uganda [21,45,46] and other countries in Sub-Saharan Africa [55,56] that further highlight the burden of infectious diseases in low-resource settings.

## Compliance with treatment guidelines

The proportion of antibiotic prescriptions that aligned with the recommended national treatment guidelines was low, 29% and 31% for outpatient and inpatient departments, respectively. Worryingly, this pattern is also observed in other regions of the country. One study investigating antibiotic prescription practices in Eastern Uganda reported only 17% of antibiotic prescriptions being compliant in accordance with the Uganda Clinical Guidelines [59]. Another point prevalence survey across 13 hospitals in Uganda found 30% compliance with Uganda Clinical Guidelines [25].

The reasons for poor compliance with the national treatment guidelines in our study are unclear. However, common reasons for poor compliance with the national treatment guidelines include poor dissemination of the guidelines, lack of diagnostics such as in microbiology and radiology, long turn-around times for laboratory results, prescribers' preferences and behaviors, and influence of pharmacies and pharmaceutical companies [58,59]. The poor compliance with national guidelines could be a contributing factor to the high prescription rate of the Watch category group of antibiotics [2]. In contrast, compliance with national guidelines in Tanzania and South Africa was high at 84% and 90.2% respectively [60,61]. The relatively high compliance with national guidelines could partly be attributed to the availability of robust national antimicrobial stewardship programs; along with strong healthcare quality improvement initiatives [62,63]. These are potentially useful strategies for Uganda to adopt, to improve compliance with treatment guidelines. KNRH is currently undergoing efforts to improve AMS

and local treatment recommendations have been developed and disseminated among the health care staff. These measures, alongside regular training, are aimed at improving AMU.

## Strengths and limitations

The findings of this study provide a basis for identifying gaps to improve antimicrobial stewardship practices at Kiruddu Hospital. The use of the standard WHO point prevalence survey (PPS) methodology allows for comparisons with future studies to monitor trends in antibiotic prescription practices, which could be used to evaluate current AMS measures and influence critical health policy-making decisions, e.g., concerning procurement. It also builds on previous efforts by the Uganda National Drug Authority and provides updated information on Antimicrobial consumption to support policy discussions at the local and national level during action plan reviews on AMR. However, we acknowledge some inherent weaknesses of this study. Firstly, the findings may not be readily generalizable to other healthcare settings since this was a single-site study at a tertiary-level healthcare facility. Secondly, the point-in-time nature of the PPS design further limits insight into seasonal patterns in antibiotic use. Thirdly, some information was incompletely documented because it was either omitted or not routinely collected. Nonetheless, we believe that the findings provide valuable lessons for improving antimicrobial stewardship practices.

## Conclusion

Antibiotic use, particularly for watch antibiotics, was high, and was associated with poor compliance to national treatment guidelines. We recommend training of prescribers on the national treatment guidelines and establishing a local antibiogram to support rational antibiotic prescribing at Kiruddu National Referral Hospital.

## Supporting information

**S1 File. Antimicrobial consumption and expenditure dataset and analysis output.**
(XLSX)

**S2 File. Outpatient department dataset and analysis output.**
(XLSX)

**S3 File. Analysis output for OPD infections and prescribed antimicrobial agents.**
(XLSX)

**S4 File. In-patient department dataset and analysis output.**
(XLSX)

## Acknowledgments

We are indebted to the Kiruddu National Referral Hospital staff and patients who participated in the study.

## Author Contributions

**Conceptualization:** Mark Kizito, Darius Owachi, Falisy Lule, Laura Jung.

**Formal analysis:** Vivian Bazanye.

**Funding acquisition:** Charles Kabugo.

**Investigation:** Mark Kizito, Falisy Lule, Vivian Bazanye.

**Methodology:** Mark Kizito, Vivian Bazanye.

**Project administration:** Darius Owachi, Charles Kabugo.

**Supervision:** Darius Owachi, Laura Jung, Ibrahim Mugerwa, Susan Nabadda, Charles Kabugo.

**Writing – original draft:** Mark Kizito, Falisy Lule.

**Writing – review & editing:** Darius Owachi, Laura Jung, Vivian Bazanye, Ibrahim Mugerwa, Susan Nabadda, Charles Kabugo.

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
