## [Decision Letter · Decision Letter 0]

26 Aug 2024

PONE-D-24-28507Antibiotic consumption and utilization at a large tertiary care level hospital in Uganda: a point prevalence survey.PLOS ONE

Dear Dr. Kizito,

Thank you for submitting your manuscript to PLOS ONE. After careful consideration, we feel that it has merit but does not fully meet PLOS ONE’s publication criteria as it currently stands. Therefore, we invite you to submit a revised version of the manuscript that addresses the points raised during the review process.

We look forward to receiving your revised manuscript.

Kind regards,

Mabel Kamweli Aworh, DVM, MPH, PhD. FCVSN

Academic Editor

PLOS ONE

Journal requirements: 1. When submitting your revision, we need you to address these additional requirements. Please ensure that your manuscript meets PLOS ONE's style requirements, including those for file naming. The PLOS ONE style templates can be found at https://journals.plos.org/plosone/s/file?id=wjVg/PLOSOne_formatting_sample_main_body.pdf and https://journals.plos.org/plosone/s/file?id=ba62/PLOSOne_formatting_sample_title_authors_affiliations.pdf. 2. We suggest you thoroughly copyedit your manuscript for language usage, spelling, and grammar. If you do not know anyone who can help you do this, you may wish to consider employing a professional scientific editing service.  The American Journal Experts (AJE) (https://www.aje.com/) is one such service that has extensive experience helping authors meet PLOS guidelines and can provide language editing, translation, manuscript formatting, and figure formatting to ensure your manuscript meets our submission guidelines. Please note that having the manuscript copyedited by AJE or any other editing services does not guarantee selection for peer review or acceptance for publication.  Upon resubmission, please provide the following: The name of the colleague or the details of the professional service that edited your manuscript A copy of your manuscript showing your changes by either highlighting them or using track changes (uploaded as a *supporting information* file) A clean copy of the edited manuscript (uploaded as the new *manuscript* file)”. 3. We note that the grant information you provided in the ‘Funding Information’ and ‘Financial Disclosure’ sections do not match.  When you resubmit, please ensure that you provide the correct grant numbers for the awards you received for your study in the ‘Funding Information’ section. 4. Thank you for stating the following financial disclosure:  [This study was funded through collaborations with the SPIDAAR program [funded by the Pfizer/Wellcome Trust foundation and the ESKAPE project [funded through the GIZ GmbH]].  Please state what role the funders took in the study.  If the funders had no role, please state: ""The funders had no role in study design, data collection and analysis, decision to publish, or preparation of the manuscript."" If this statement is not correct you must amend it as needed. Please include this amended Role of Funder statement in your cover letter; we will change the online submission form on your behalf. 5. We note that your Data Availability Statement is currently as follows: [All relevant data are within the manuscript and its Supporting Information files.] Please confirm at this time whether or not your submission contains all raw data required to replicate the results of your study. Authors must share the “minimal data set” for their submission. PLOS defines the minimal data set to consist of the data required to replicate all study findings reported in the article, as well as related metadata and methods (https://journals.plos.org/plosone/s/data-availability#loc-minimal-data-set-definition). For example, authors should submit the following data: - The values behind the means, standard deviations and other measures reported;- The values used to build graphs;- The points extracted from images for analysis. Authors do not need to submit their entire data set if only a portion of the data was used in the reported study. If your submission does not contain these data, please either upload them as Supporting Information files or deposit them to a stable, public repository and provide us with the relevant URLs, DOIs, or accession numbers. For a list of recommended repositories, please see https://journals.plos.org/plosone/s/recommended-repositories. If there are ethical or legal restrictions on sharing a de-identified data set, please explain them in detail (e.g., data contain potentially sensitive information, data are owned by a third-party organization, etc.) and who has imposed them (e.g., an ethics committee). Please also provide contact information for a data access committee, ethics committee, or other institutional body to which data requests may be sent. If data are owned by a third party, please indicate how others may request data access.

Reviewers' comments:

Reviewer's Responses to Questions

**Comments to the Author**

1. Is the manuscript technically sound, and do the data support the conclusions?

Reviewer #1: Yes

Reviewer #2: No

Reviewer #3: Yes

2. Has the statistical analysis been performed appropriately and rigorously? 

Reviewer #1: Yes

Reviewer #2: No

Reviewer #3: Yes

3. Have the authors made all data underlying the findings in their manuscript fully available?

Reviewer #1: Yes

Reviewer #2: Yes

Reviewer #3: Yes

4. Is the manuscript presented in an intelligible fashion and written in standard English?

Reviewer #1: Yes

Reviewer #2: Yes

Reviewer #3: Yes

5. Review Comments to the Author

Reviewer #1: The authors did a very good job presenting thier findings in this manuscript. I must commend their efforts.

However, there are some areas that may need some clarification and refinement. These are highlighted below and in the attached manuscript

Abstract

line 35: What does OPD mean? write in full for the first use.

line 36: The authors said they used a "single-day consecutive sampling". If they selected all patients then this cannot be called sampling. I suggest you avoid the word "sampling" and just say you selected ALL inpatients.

line 36: What does IPD mean? please write in full for the first use.

Main Body

INTRODUCTION

lines 56 - 86: Your introduction provides adequate background information on antibiotic resistance of global relevance, however, not much is said about the regional relevance before talking specifically about the setting in Uganda. I encourage you to include this information.

lines 74 - 78: Your introduction needs to provide clear issues related to antibiotic consumption and utilization in Uganda or in a similar setting and also Is there is no clear explanation of why antibiotic stewardship is important in tertiary care settings in Uganda. In other words, what is the gap in literature that this study aims to fill? please add these to the introduction

line 80: Do you mean "Challenges"? "Challenges" seem to be a better word here than "importance"

METHODS

Lines 89 - 101: It is important to include information that helps to contextualize the study setting.

please include:

1. Information on the hospital's patient demographics

2. The location of the hospital in terms latitude and longitude, geographical setting (rural or urban)

3. Are there any relevant local factors that might influence antibiotic usage, such as endemic diseases, public

health infrastructure, or local antimicrobial resistance patterns?

4. There is zero detail about the hospital’s pharmacy services, laboratory capabilities, and infection control

practices?

5. How about the the availability of antibiotics in the facility? Are there any seasonal or temporal factors in the

area that could impact antibiotic usage?

all these are important to give proper context to your study and findings.

line 109: do you mean 1st of July to 30th June? Please be specific with dates

Line 124: can you clarify how you arrived at this value of 200? please briefly include the parameters used to arrive at this value.

line 125: In the actual sense you did not have anything to do with the pateints directly so i would not say "recruited". you only worked with patient records is that correct? Please stay away from using the words recruited, enrolled e.t.c

Line 124: Can you clarify how you arrived at the sampling interval of 44? There is also no mention of your sample frame? Is it that you had a long list of all the OPD patients in a sequence of how they were recorded? please briefly explain these in clear terms so the readers can understand your methodology.

Did you have any inclusion or exclusion criteria??

line 132: Please include the version of ODK as well as the reference.

Can you also give more details about this tool that you used to collect the data? was this a proforma that was adopted from the WHO PPS or you adapted this tool to suit your study? what is the validity of this tool? what is the structure of this tool? did it have a section for sociodemographic characteristics of the patient? Little is known about the tool you used. please clarify or give more details.

Line 146: How did you ensure data security? It will be a good addition to your ethical considerations

RESULT

line 156: It is incorrect to say patients were enrolled when in essence only patient records were reviewed. It is my understanding that you had absolutely no contact with patients. Patient enrollment usually refers to the process of actively recruiting and consenting individuals to participate in a study, where they are directly involved through surveys or clinical trials.

Line 158: Why are you using median and IQR instead of Mean and Standard deviation? Is there something more about the data you need to tell us?

Line 160: Please maintain consistency in presentation of your results for univariate analysis of categorocal variables. In one hand you present just the frequency and percent e.g "Females comprised 130 (61.6%)" and in another hand you present a proportion and the percent in parenthesis e.g outpatient (150/183, 160 82%) and within the same paragraph you change to just frequency and percent in parenthesis e.g "diabetes mellitus (7, 14.9%)". please chose one method and stick with it to avoid confusion

Line 164: why use the term "Baseline" here? Please clarify

Table 1: What do we mean by "OPD(%)" as a column header? This is not the same as frequency or is this another measure?? same for the second column

Table 1: Does this data fit into this column??? the column header is saying OPD(%). i dont think this is correct. please put your data properly in the correct table structure or present as prose

Line 187: Do you mean you reviewed patients or patient records???

DISCUSSION

Lines 314 - 316: rephrase this sentence. it appears to have a syntax problem. Maybe change the word "Undergoes" to "undergoing" to match the present continuous tense. "Amongst" can also be changed to "among" for a more straightforward usage.

Lines 317 - 330: In this study, there is no mention of incomplete records or data on antibiotic use was missing. considering that you used hospital records which in many settings are incomplete. How did you deal with such situations as this is usually a limitation and could lead to inaccuracies in the survey results.

Lines 333 - 334: This recommendation appears general. Tailor your recommendations to specific audiences (e.g., healthcare providers, policymakers, hospital administrators) so they know what actions they can take.

Reviewer #2: Thank you for inviting me to review this manuscript on Antibiotic consumption and utilization at a large tertiary care level hospital in Uganda: a point prevalence survey.

Please find below, my comments and quality-improvement recommendations to the authors according to line numbers.

51 and 52: The inpatient indications for antibiotic therapy are listed as sepsis - 12.2%, respiratory infections - 18.3%, burn wounds - 14.1%, and gastrointestinal infections - 14.1%. These make up 74.7%. You need to account for the remaining 25.3% even if it is “others”.

53: Your study concludes that the prescription of “watch” antibiotics was high. Is this observation based on an objective of your study? Did you want to determine which type/group/class of antibiotics was mostly prescribed?

67: Please change from "misuse and -use" to "use and misuse" and note that self-medication and antibiotic over-prescription are examples of use and misuse of antibiotics, not separate issues.

69: Please state what the health agenda of the June 2021 G7 meeting was. It is an important detail to include especially if you want to use it to validate your work.

75 and 76: The high rates reported in the hospital "were" not "was".

79 and 80: Please state this clearly. Does it pose significant public health concerns of great socio-economic importance? Or what did you want us to know?

85: Your study aims to determine what exactly? Is it the prevalence of antimicrobial resistance or the rate of antibiotic prescription and/or misuse? What are the objectives of your study? These would have guided your write-up. Did you want to simply show high rates of antibiotic prescriptions? Or did you want to compare antibiotic resistance/prescription to the findings in other countries in East Africa, Africa as a whole, or outside Africa? Or did you want to assess the risk of developing AMRs in Uganda based on the records from Kiruddu Hospital?

90 to 95: Please rewrite this paragraph. There are too many redundant words here. Seeing that the study was not exactly about Kiruddu HRH, you did not need to describe it to the tiniest detail.

97: Please write SPIDAAR and ESKAPE in full at first mention with the acronyms in parentheses. You may then use the acronyms in subsequent mentions.

104: What makes up "all data"? All data regarding antibiotic prescription, adverse effects, patient compliance or reasons for non-compliance? Please clarify which data you collected and the exceptions you had.

160: Are the comorbidities significant? Did they in some way encourage or deter antibiotic use and/or misuse? Or did they contribute to antibiotic resistance? Please clarify this statement so that we can easily understand your point.

166: Why the in-depth financial analysis? Is it an objective of your work to show the financial implications of antibiotic use/misuse/resistance? This takes us back to the need for you to identify and clearly state your objectives. They are the road signs to guide your work.

173: It cannot be "most consumed" individual antibiotic and be so many. Please rewrite this.

222: “The total DDD per thousand inhabitants per day ‘was’ (not were) 6.05.”

273 to 275: Both sentences mean the same thing. You need only one

276: You should give a brief description of antibiograms so that we can understand why you think they can help to reduce indiscriminate antibiotic use and the development of AMRs.

303 to 304: It is said that common things occur commonly. Is it possible for the common reasons for poor compliance with national treatment guidelines to apply to your study?

333 - 334: “Establishment of robust antimicrobial stewardship measures could help improve and monitor the rational prescription of antibiotics.” Measures like what? You mentioned antibiotic stewardship programs several times but you neither explained these in practical terms nor did you give existing examples that can be copied by the Ugandan health administration.

Finally, what is the WHO’s AWaRE classification of antibiotics and how are the classes different from one another? If you found that most of the antibiotics in your study belonged to the "watch" class, what does this mean in terms of indications for use, tendency for self-medication, level of patient compliance, and likelihood of developing resistance?

Reviewer #3: I found the manuscript well written, good research question and the outcome was intelligently communicated. However, the authors were silent about the accessibility of POM antibiotics over the counter by the populace without any prescription as seen in most countries in the Sub-Saharan African region. These could also be a silent factor contributing to AMR because antibiotics are not easily accessible by people in western world.

Minor corrections observed are highlighted in the manuscript.

6. PLOS authors have the option to publish the peer review history of their article (what does this mean?). If published, this will include your full peer review and any attached files.

Reviewer #1: **Yes: **Abdulhakeem Abayomi Olorukooba

Reviewer #2: **Yes: **Oluwafolayemi Doyeni

Reviewer #3: No

---

## [Author Response · Author response to Decision Letter 0]

9 Oct 2024

Here is a point by point response to the reviewers' comments and concerns

Reviewer #1

The authors did a very good job presenting their findings in this manuscript. I must commend their efforts. However, there are some areas that may need some clarification and refinement. These are highlighted below and in the attached manuscript

 Abstract

 line 35: What does OPD mean? write in full for the first use.

Answer: Thank you for pointing this out. It has been addressed in line 27-28.

 line 36: The authors said they used a "single-day consecutive sampling". If they selected all patients then this cannot be called sampling. I suggest you avoid the word "sampling" and just say you selected ALL inpatients.

Answer: This has been changed to “selected all inpatients” (line 27-28).

 line 36: What does IPD mean? please write in full for the first use.

Answer: The term IPD has been removed and replaced with inpatient department (line 28). 

 Main Body

 INTRODUCTION

 lines 56 - 86: Your introduction provides adequate background information on antibiotic resistance of global relevance, however, not much is said about the regional relevance before talking specifically about the setting in Uganda. I encourage you to include this information.

Answer: This has been addressed (Line 56-57)

 lines 74 - 78: Your introduction needs to provide clear issues related to antibiotic consumption and utilization in Uganda or in a similar setting and also Is there is no clear explanation of why antibiotic stewardship is important in tertiary care settings in Uganda. In other words, what is the gap in literature that this study aims to fill? please add these to the introduction

Answer: We laid out the currently available data as well as the gaps, especially with regard to tertiary care centers, where a higher use of “watch” and “reserve” category antibiotics is to be expected (line 72-80). 

 line 80: Do you mean "Challenges"? "Challenges" seem to be a better word here than "importance"

Answer: Thank you for your suggestion. The word importance has been replaced with challenges (line 87)

 METHODS

 Lines 89 - 101: It is important to include information that helps to contextualize the study setting.

 please include:

 1. Information on the hospital's patient demographics - updated in lines 123-124

 2. The location of the hospital in terms latitude and longitude, geographical setting (rural or urban) - this has been updated in line 119

 3. Are there any relevant local factors that might influence antibiotic usage, such as endemic diseases, public health infrastructure, or local antimicrobial resistance patterns?

 4. There is zero detail about the hospital’s pharmacy services, laboratory capabilities, and infection control practices?

 5. How about the the availability of antibiotics in the facility? Are there any seasonal or temporal factors in the area that could impact antibiotic usage?

all these are important to give proper context to your study and findings.

Answer: Thank you for raising these concerns. All these have been addressed (lines 99-118)

 line 109: do you mean 1st of July to 30th June? Please be specific with dates

Answer: This has been addressed. The dates are 1st July 2021 through 30th June 2022 (line 132).

 Line 124: can you clarify how you arrived at this value of 200? please briefly include the parameters used to arrive at this value. 

Answer: This value is from the WHO guidance on "How to investigate Medicines Use in Health Facilities". In the chapter on sample size, the guidance is to consider 100 patient encounters (prescriptions). We consider 200 prescriptions (regardless of whether they got antibiotics) to ensure that the data set for those with antibiotics includes at least 100 prescriptions given documented antibiotic prevalence rates.

 line 125: In the actual sense you did not have anything to do with the patients directly so i would not say "recruited". you only worked with patient records is that correct? Please stay away from using the words recruited, enrolled e.t.c

Answer: Thank you for pointing this out. The word recruited has been replaced with the term selected.

 Line 124: Can you clarify how you arrived at the sampling interval of 44? There is also no mention of your sample frame? Is it that you had a long list of all the OPD patients in a sequence of how they were recorded? please briefly explain these in clear terms so the readers can understand your methodology. Did you have any inclusion or exclusion criteria??

Answer: The information has been elaborated upon in lines 150-153. There was no specific inclusion or exclusion criteria.

 line 132: Please include the version of ODK as well as the reference.

 Can you also give more details about this tool that you used to collect the data? was this a proforma that was adopted from the WHO PPS or you adapted this tool to suit your study? what is the validity of this tool? what is the structure of this tool? did it have a section for sociodemographic characteristics of the patient? Little is known about the tool you used. please clarify or give more details.

Answer: We used ODK Collect v2022.4.3 tool designed based on the WHO point prevalence survey methodology as it is. No variables were removed or changed to suit our context. (lines 160- 162)

 Line 146: How did you ensure data security? It will be a good addition to your ethical considerations

Answer: All data were stored at a secure server located at the Infectious Diseases Institute, Makerere University (Lines 181- 182)

 RESULT

 line 156: It is incorrect to say patients were enrolled when in essence only patient records were reviewed. It is my understanding that you had absolutely no contact with patients. Patient enrollment usually refers to the process of actively recruiting and consenting individuals to participate in a study, where they are directly involved through surveys or clinical trials.

Answer: Thank you once again for pointing this out. The term enrolled has been removed (line 185)

 Line 158: Why are you using median and IQR instead of Mean and Standard deviation? Is there something more about the data you need to tell us?

Answer: We used median and IQR instead of mean and standard deviation because the age data was right skewed.

 Line 160: Please maintain consistency in presentation of your results for univariate analysis of categorical variables. In one hand you present just the frequency and percent e.g "Females comprised 130 (61.6%)" and in another hand you present a proportion and the percent in parenthesis e.g outpatient (150/183, 160 82%) and within the same paragraph you change to just frequency and percent in parenthesis e.g "diabetes mellitus (7, 14.9%)". please choose one method and stick with it to avoid confusion

Answer: This has been addressed to reflect frequency and percentage.

 Line 164: why use the term "Baseline" here? Please clarify

Answer: Thank you for pointing this out. There is no need to use the term baseline since no intervention was applied. The term baseline has been removed (Line 193)

 Table 1: What do we mean by "OPD(%)" as a column header? This is not the same as frequency or is this another measure?? same for the second column. 

 Table 1: Does this data fit into this column??? the column header is saying OPD(%). i don't think this is correct. please put your data properly in the correct table structure or present as prose

Answer: OPD (Outpatient department) and IPD (inpatient department) refer to the different settings where the study was conducted. For each variable, we segregated according to the setting (whether OPD or IPD) and presented the data in absolute figures (n) with their corresponding proportions in brackets.

 Line 187: Do you mean you reviewed patients or patient records??? Answer: We reviewed patient records and this has been corrected (Line 215)

 DISCUSSION

 Lines 314 - 316: rephrase this sentence. it appears to have a syntax problem. Maybe change the word "Undergoes" to "undergoing" to match the present continuous tense. "Amongst" can also be changed to "among" for a more straightforward usage.

Answer: Thank you for these suggestions. The word undergoes has been replaced with undergoing and amongst replaced with among (lines 352 - 354)

 Lines 317 - 330: In this study, there is no mention of incomplete records or data on antibiotic use was missing. considering that you used hospital records which in many settings are incomplete. How did you deal with such situations as this is usually a limitation and could lead to inaccuracies in the survey results.

Answer: Thank you so much. This limitation has been included. some information was incompletely documented because it was either omitted or not routinely collected (Lines 366 - 367)

 Lines 333 - 334: This recommendation appears general. Tailor your recommendations to specific audiences (e.g., healthcare providers, policymakers, hospital administrators) so they know what actions they can take.

Answer: This has been addressed. We recommend training of prescribers on the national treatment guidelines and establishing a local antibiogram to support rational antibiotic prescribing at Kiruddu National Referral Hospital (Lines 372 - 374)

Reviewer #2

Thank you for inviting me to review this manuscript on Antibiotic consumption and utilization at a large tertiary care level hospital in Uganda: a point prevalence survey.

 Please find below, my comments and quality-improvement recommendations to the authors according to line numbers.

 51 and 52: The inpatient indications for antibiotic therapy are listed as sepsis - 12.2%, respiratory infections - 18.3%, burn wounds - 14.1%, and gastrointestinal infections - 14.1%. These make up 74.7%. You need to account for the remaining 25.3% even if it is “others”.

Answer: Thank you for pointing this out. However, respiratory infections accounted for 38.1% not 18.3% as stated bringing the total to 91.3%. Medical prophylaxis accounted for the remaining 8.7% and this has been included in the abstract (Lines 41 - 42).

 53: Your study concludes that the prescription of “watch” antibiotics was high. Is this observation based on an objective of your study? Did you want to determine which type/group/class of antibiotics was mostly prescribed?

Answer: The study was designed to broadly determine the antimicrobial consumption and utilization of Kiruddu National Referral Hospital for financial year 2021- 2022 including but not limited to the types/group/ classes of antibiotics prescribed. Other specific objectives included determining the indications for antibiotic use, expenditure on antibiotics, and daily defined doses (DDD) for antibiotics purchased in that financial year. 

 67: Please change from "misuse and -use" to "use and misuse" and note that self-medication and antibiotic over-prescription are examples of use and misuse of antibiotics, not separate issues.

Answer: Thank you for these suggestions. We have incorporated them into the manuscript (lines 63- 64)

 69: Please state what the health agenda of the June 2021 G7 meeting was. It is an important detail to include especially if you want to use it to validate your work.

Answer: This has been addressed (Lines 67- 71)

 75 and 76: The high rates reported in the hospital "were" not "was".

Answer: This has been addressed.

 79 and 80: Please state this clearly. Does it pose significant public health concerns of great socio-economic importance? Or what did you want us to know?

Answer: AMR poses significant public health and socio-economic challenges not importance. This has been addressed in line 87

 85: Your study aims to determine what exactly? Is it the prevalence of antimicrobial resistance or the rate of antibiotic prescription and/or misuse? What are the objectives of your study? These would have guided your write-up. Did you want to simply show high rates of antibiotic prescriptions? Or did you want to compare antibiotic resistance/prescription to the findings in other countries in East Africa, Africa as a whole, or outside Africa? Or did you want to assess the risk of developing AMRs in Uganda based on the records from Kiruddu Hospital?

Answer: The study was designed to broadly determine the antimicrobial consumption and utilization of Kiruddu National Referral Hospital for financial year 2021- 2022 including but not limited to the types/group/ classes of antibiotics prescribed. Other specific objectives included determining the indications for antibiotic use, expenditure on antibiotics, and daily defined doses (DDD) for antibiotics purchased in that financial year.

 90 to 95: Please rewrite this paragraph. There are too many redundant words here. Seeing that the study was not exactly about Kiruddu HRH, you did not need to describe it to the tiniest detail.

Answer: The reviewer requests with this respect to section “study design and setting” are opposing each other. We therefore provided a suggestion but would leave it to the editors to decide the level of detail in line with the journal's requirements. 

 97: Please write SPIDAAR and ESKAPE in full at first mention with the acronyms in parentheses. You may then use the acronyms in subsequent mentions.

Answer: This information has been removed since it is not relevant.

 104: What makes up "all data"? All data regarding antibiotic prescription, adverse effects, patient compliance or reasons for non-compliance? Please clarify which data you collected and the exceptions you had.

Answer: We collected data on the types of antibiotics, their quantities and indications for antibiotic use. We did not collect data on adverse effects of antibiotics. We go ahead to specify the data that was collected in the next paragraphs (Lines 132- 133 & 159- 171). 

 160: Are the comorbidities significant? Did they in some way encourage or deter antibiotic use and/or misuse? Or did they contribute to antibiotic resistance? Please clarify this statement so that we can easily understand your point.

Answer: Thank you for this concern. Yes, comorbidities affect the use of antibiotics. Some patients with certain comorbidities are likely to receive antibiotic prescriptions. Based on this, we believe it is important to show the number of patients that had various comorbidities.

 166: Why the in-depth financial analysis? Is it an objective of your work to show the financial implications of antibiotic use/misuse/resistance? This takes us back to the need for you to identify and clearly state your objectives. They are the road signs to guide your work.

Answer: Under antimicrobial consumption, we specifically set out to determine the expenditure incurred in purchase of all antibiotics in the financial year 2021-2022. This was one of our objectives.

 173: It cannot be "most consumed" individual antibiotic and be so many. Please rewrite this.

Answer: This has been rewritten (Lines 201- 204)

 222: “The total DDD per thousand inhabitants per day ‘was’ (not were) 6.05.”

Answer: This has been corrected (Line 249-250)

 273 to 275: Both sentences mean the same thing. You need only one

Answer: We agree and deleted the first sentence.

 276: You should give a brief description of antibiograms so that we can understand why you think they can help to reduce indiscriminate antibiotic use and the development of AMRs.

Answer: This has been addressed. Lines 315- 318)

 303 to 304: It is said that common things occur commonly. Is it possible for the common reasons for poor compliance with national treatment guidelines to apply to your study?

Answer: True, the common reasons for poor compliance with national treatment guidelines apply to my study and these were outlined in lines 341- 345.

 333 - 334: “Establishment of robust antimicrobial stewardship measures could help improve and monitor the rational prescription of antibiotics.” Measures like what? You mentioned antibiotic stewardship programs several times but you neither explained these in practical terms nor did you give existing examples that can be copied by the Ugandan health administration.

Answer: Thank you for pointing this out. This has been addressed in lines 372- 374.

 Finally, what is the WHO’s AWaRE classification of antibiotics and how are the

---

## [Decision Letter · Decision Letter 1]

29 Oct 2024

Antibiotic consumption and utilization at a large tertiary care level hospital in Uganda: a point prevalence survey.

PONE-D-24-28507R1

Dear Dr. Kizito,

We’re pleased to inform you that your manuscript has been judged scientifically suitable for publication and will be formally accepted for publication once it meets all outstanding technical requirements.

Kind regards,

Mabel Kamweli Aworh, DVM, MPH, PhD. FCVSN

Academic Editor

PLOS ONE

Additional Editor Comments (optional):

Reviewers' comments:

Reviewer's Responses to Questions

**Comments to the Author**

1. If the authors have adequately addressed your comments raised in a previous round of review and you feel that this manuscript is now acceptable for publication, you may indicate that here to bypass the “Comments to the Author” section, enter your conflict of interest statement in the “Confidential to Editor” section, and submit your "Accept" recommendation.

Reviewer #1: All comments have been addressed

Reviewer #2: All comments have been addressed

Reviewer #3: All comments have been addressed

2. Is the manuscript technically sound, and do the data support the conclusions?

Reviewer #1: Yes

Reviewer #2: (No Response)

Reviewer #3: Yes

3. Has the statistical analysis been performed appropriately and rigorously? 

Reviewer #1: Yes

Reviewer #2: (No Response)

Reviewer #3: Yes

4. Have the authors made all data underlying the findings in their manuscript fully available?

Reviewer #1: Yes

Reviewer #2: (No Response)

Reviewer #3: Yes

5. Is the manuscript presented in an intelligible fashion and written in standard English?

Reviewer #1: Yes

Reviewer #2: (No Response)

Reviewer #3: Yes

6. Review Comments to the Author

Reviewer #1: The Authors did a great job attending to the comments. All my comments have been satisfactorily addressed.

Reviewer #2: (No Response)

Reviewer #3: The authors have attended to all the concerns I raised earlier, and the manuscript is intelligently written

7. PLOS authors have the option to publish the peer review history of their article (what does this mean?). If published, this will include your full peer review and any attached files.

Reviewer #1: **Yes: **Abdulhakeem Abayomi Olorukooba

Reviewer #2: **Yes: **Dr. Oluwafolayemi Doyeni

Reviewer #3: No

---

## [Editor Report · Acceptance letter]

2 Jan 2025

PONE-D-24-28507R1 

PLOS ONE

Dear Dr. Kizito, 

I'm pleased to inform you that your manuscript has been deemed suitable for publication in PLOS ONE. Congratulations! Your manuscript is now being handed over to our production team.

Kind regards, 

on behalf of

Dr. Mabel Kamweli Aworh 

Academic Editor

PLOS ONE